# Synthesis, Solution, and Solid State Properties of Homological Dialkylated Naphthalene Diimides—A Systematic Review of Molecules for Next-Generation Organic Electronics

**DOI:** 10.3390/molecules28072940

**Published:** 2023-03-25

**Authors:** Dorota Chlebosz, Waldemar Goldeman, Krzysztof Janus, Michał Szuster, Adam Kiersnowski

**Affiliations:** 1Department of Physical and Quantum Chemistry, Wroclaw University of Science and Technology, Wybrzeże Wyspiańskiego 27, 50-370 Wroclaw, Poland; 2The Leibniz Institute of Polymer Research, Hohe Strasse 6, D-01069 Dresden, Germany; 3Department of Medicinal and Organic Chemistry, Wroclaw University of Science and Technology, Wybrzeże Wyspiańskiego 27, 50-370 Wroclaw, Poland

**Keywords:** naphthalene diimide, spectroscopy, thermal analysis, morphology, solubility, field-effect mobility, odd–even parity effect

## Abstract

This systematic study aimed at finding a correlation between molecular structure, solubility, self-assembly, and electronic properties of a homological series of *N*-alkylated naphthalene diimides (NDIs). NDIs are known for their n-type carrier mobility and, therefore, have potential in the field of organic electronics, photovoltaics, and sensors. For the purpose of this study, nine symmetrical *N*,*N*′-dialkylated naphthalene diimides (NDIC3-NDIC11) were synthesized in the reaction of 1,4,5,8-naphthalenetetracarboxylic dianhydride with alkylamines ranging from propyl- to undecyl-. The NDIs were characterized by spectroscopic (NMR, UV-Vis, FTIR), microscopic, and thermal methods (TGA and DSC), and X-ray diffraction (XRD). Our experimental study, extensively referring to findings reported in the literature, indicated that the NDIs revealed specific trends in spectroscopic and thermal properties as well as solubility and crystal morphology. The solubility in good solvents (chloroform, toluene, dichlorobenzene) was found to be the highest for the NDIs substituted with the medium-length alkyl chains (NDIC5–NDIC8). Systematic FTIR and XRD studies unraveled a distinct parity effect related to the packing of NDI molecules with odd or even numbers of methylene groups in the alkyl substituents. The NDIs with an even number of methylene groups in the alkyl substituents revealed low-symmetry (P$\overset{-}{1}$) triclinic packing, whereas those with an odd number of carbon atoms were generally monoclinic with P2_1_/c symmetry. The odd–even parity effect also manifested itself in the overlapping of the NDIs’ aromatic cores and, hence, the π-π stacking distance (d_π-π_). The odd-numbered NDIs generally revealed slightly smaller d_π-π_ values then the even-numbered ones. Testing the NDIs using standardized field-effect transistors and unified procedures revealed that the n-type mobility in NDIC6, NDIC7, and NDIC8 was 10- to 30-fold higher than for the NDIs with shorter or longer alkyl substituents. Our experimental results indicate that *N,N′*-alkylated NDIs reveal an optimum range of alkyl chain length in terms of solution processability and charge transport properties.

## 1. Introduction

The *N*,*N*′-dialkylated naphthalene diimides (NDIs) represent the group of aromatic diimides also referred to as rylene dyes (Figure 1). The NDIs and their larger-core counterparts, perylenediimides (PDIs, Figure 1), are the most extensively studied rylene dyes. NDIs are molecules with a remarkable planar structure built up of fused benzene rings (Figure 1). They exhibit high chemical, thermal, and oxidative stability. The high stability in conjunction with their redox activity [1], spectroscopic [2,3], and electronic properties [4,5] make NDIs interesting for materials scientists and supramolecular chemists. NDIs reveal higher field-effect electron mobility than pyromellitic diimides (PyDI, Figure 1) [6,7]. The electron mobility determined in conventional, NDI thin film-based field-effect transistors was found comparable to single crystal-based PDIs [7,8]. Unlike the PDIs, however, the NDIs are colorless, which makes them useful for the fabrication of transparent, flexible electronics. Moreover, NDIs exhibit significantly higher solubility than that observed in PDIs [9].

Historically, NDIs were first mentioned as early as in the 1930s in the patents and articles on their use as various dyes [10,11,12,13,14]. The use of NDIs as dyes continued until the mid-1970s. At the turn of the 1970s and 1980s, the NDIs attracted interest from biochemists due to their ability to achieve intercalative binding of DNA [15]. Later in the 1980s, NDIs gained the interest of materials scientists due to their potential in organic field-effect transistors (OFETs), electroluminescent diodes (OLEDs), and other, also flexible, electronic devices, energy storage materials, and sensorics. NDI-based OFETs typically reveal electron mobility oscillating around 0.2 cm^2^ × V^−1^ × s^−1^. The OFET based on the *N,N′*-dicyclohexyl-2,6-naphthalene NDI revealed electron mobility reaching as high as 6 cm^2^ × V^−1^ × s^−1^ [7,16,17,18]. NDIs have been successfully used to enhance the electronic performance of polymer-based OFETs [19,20,21]. Lithium-ion batteries with NDI-based electrodes revealed discharge potentials ranging from 2.3 to 2.9 V [22]. NDIs were successfully used as acceptors in organic light-emitting diodes (OLEDs) [23] or in flexible photovoltaic cells (OPV) with power conversion efficiency of 7.65% [24]. NDIs were also used as molecular sensors, playing the role of acceptors in host–guest-type complexes or as an aromatic electron acceptor in synthetic ion channels [25,26]. 

Interest in the synthesis and applications of NDIs is still on the rise despite decades of preceding research [27,28,29,30,31]. There are novel applications reported in the fields of organic electronics, modern power sources, or power storage units [29,30,32]. NDIs are still studied for their photophysical properties [3]. Nonetheless, in the literature, consistent systematic investigations of homological series of NDIs are still relatively rare [33]. Such a survey for symmetrical NDIs is necessary in order to support the engineering of NDI-based materials. The research papers typically focus on selected properties of arbitrary selected NDIs. In our previous paper, we have already faced the problem of selecting NDI derivatives for specific purposes in the fabrication of OFETs [21]. Motivated by the issues we have experienced in our previous research, herein, we extensively survey—both in terms of literature review and experimentally—the physical properties of homological series of symmetrically *N,N′*-dialkylated NDIs. The NDIs in this study were synthesized, purified, and analyzed in a unified, consistent way which, therefore, enabled their direct comparison. In addition to the experimental data, for the sake of comparison, here, we also provide and discuss experimental results previously reported in the literature. This extensive study unveils the characteristic trends in the physical properties of *N,N′*-dialkylated NDIs. We demonstrate here that NDIs substituted with moderately long alkyl chains reveal the highest solubility in good solvents and the highest charge carrier mobility. Calorimetric analysis of the full array of n-propyl- through n-undecyl-substituted NDIs indicated that a stepwise extending of the alkyl substituents leads to a gradual fading of aromatic-related and increase in alkyl-related features with a relatively well-distinguished range of molecules with intermediate properties. Based on such observations, we demonstrate and discuss the patterns of properties characteristic of a homological series of *N,N′*-dialkylated NDIs. 

## 2. Results and Discussion

### 2.1. Remarks on Synthesis: Towards Increasing the Synthesis Scale

The general synthetic route of NDIs is shown in Figure 2. 

The synthetic protocols of NDIs reported in the literature involve heating of 1,4,5,8-naphthalenetetracarboxylic dianhydride with an amine in a solvent at elevated temperature. Depending on the temperature (ranging typically from 100 to 220 °C) and reactivity of the amine, the reaction lasted from a few to 48 h [34,35,36,37]. After the reaction, the crude product was usually precipitated with methanol or ethanol, and separated by filtration. The separated crude product was purified by column chromatography, sublimation, or recrystallization. 

Most of the synthetic protocols are typically based on the 1–4 mmol scale [34,35,38,39,40]. Due to extensive analytical needs in this study, larger quantities were required. That motivated a search for an alternative synthetic route permitting at least 4 g (7–11 mmol) scale. Experimental verification of the reported synthetic protocols indicated that the most scale-limiting factors were the viscosity of the reaction medium and the efficiency of final NDI isolation. Typically, DMF [34,35], quinoline [41,42], or pyridine [22,36] were used as reaction media. The imidazole-toluene mixture was found a less reported reaction medium [37,43]. The use of molten imidazole rather than quinoline as a reaction medium was first reported by Langhals in the synthesis of PDIs [43]. Both the imidazole and quinoline were found to increase the yield, especially in the case of less reactive, sterically hindered amines. Both imidazole and quinoline can increase the solubility of the anhydride (Figure 2). The use of imidazole permits decreasing the reaction temperature from approx. 180–230 °C (typical range for quinoline) down to 100–130 °C [43], which points towards a catalytic influence of imidazole on the reaction. Despite an unambiguously positive impact on the reaction, the role of imidazole was not finally clarified. Toluene lowers viscosity and, therefore, facilitates stirring and prevents splashing of the reaction mixture [8,44,45,46]. Moreover, the imidazole–toluene mixture facilitates NDI isolation: toluene—unlike the aforementioned DMF, quinoline, or pyridine—can be easily evaporated. Imidazole can be effectively removed during washing with methanol. The yields for 5 mmol-scale synthesis in the imidazole–toluene mixture were reported to reach 80%. These advantages underpinned the decision to choose the imidazole–toluene mixture in our study. Because of the reaction scale, conventional multiple washing on a filter funnel was replaced with continuous washing with methanol in the Soxhlet apparatus. Additional purification of NDIC3 and NDIC4 (short-column chromatography/sublimation) was necessary due to contamination of NDIs with monoimides resulting probably from the low boiling point of n-propylamine and n-butylamine (see Table 1); evaporation of the amines probably caused a variation in the reagents’ ratio over time. Final NDIs were obtained with 35–88% yields (Table 1). The highest yields, exceeding 80%, were reached for NDIC5, NDIC6, and NDIC9. The yields were found, to some extent, correlated with the amines’ boiling points. In principle, the amines with boiling points in the range from 100 to 220 °C permitted the higher yields (Table 1). 

### 2.2. UV-Vis Spectroscopy: Solubility of the NDIs

The UV-Vis spectra of NDIs in 0.1 mM chloroform solutions, their description, and molar absorptivity (ε) data are presented in Appendix A. Generally, all NDIs showed the same characteristic UV-Vis absorption features, regardless of the length of the alkyl chains. In the UV-Vis spectra of the NDIs, three characteristic sharp absorption bands at 342, 360, and 380 nm were identified. Such bands are characteristic of symmetrical NDIs and are related to the π-π* *z*-polarized transition [47,48]. The position of each band is practically constant irrespective of the alkyl chains’ length. Such absorption characteristics provide the perfect opportunity to quantify the solubility of NDIs in different solvents based on an analysis of the UV-Vis spectra. 

It is generally known that NDIs are soluble in the lipophilic solvents like chloroform, dichloromethane, or toluene, or in polar aprotic solvents like DMSO or DMF [49,50,51,52]. The solubility of NDIs in cyclohexane, n-hexane, or n-heptane is significantly lower. In our study, we tested and verified the solubility of NDIs in solvents commonly used in the fabrication of organic electronic films [5,51,53]. The solubility of NDIs was, therefore, tested in chloroform, 1,2-dichlorobenzene, and toluene—good solvents for NDIs, as well as n-heptane—a poor solvent [49,53,54]. Our solubility tests indicated that the solubility of NDIC5 through NDIC8 and of NDIC10 in chloroform exceeded 100 mg/mL. The solubility of the NDIC3, NDIC4, NDIC9, and NDIC11 in chloroform was 46.9 ± 1.5, 65.5 ± 1.6, 47 ± 1.0, and 15.15 ± 0.6 mg/mL, respectively. Considering applications of NDIs in, for instance, printed organic electronics, that solubility in the case of all NDIs exceeds usual needs since typically used concentrations oscillate within 2–20 mg/mL [5,51,53]. Conversely, the solubility of NDIs in n-heptane (Figure 3 and Appendix A) was two orders of magnitude lower than in chloroform. Generally, solubility in n-heptane was found to increase with increasing alkyl chain length. The solubility of NDIC11 was found approx. 6-fold higher than that of NDIC3. Plotting the solubility against alkyl chain length (Figure 3a) seems to reveal two distinct trends: a sharp power-like increase from NDIC3 through NDIC6, and a quasi-linear trend starting from a local minimum observed for NDIC7 up to NDIC11. 

The solubility of NDIs in toluene and 1,2-dichlorobenzene revealed different trends. In both these cases, maximum solubility was observed for NDIC6 (Figure 3b,c and Appendix A). NDIC5 through NDIC8 reveal the best solubility in toluene and 1,2-dichlorobenzene, hence distinguishing them in the whole range of NDIs. Extending the alkyl chains beyond 8 carbon atoms significantly reduced the solubility. Despite the NDIC3 has the shortest alkyl substituents among the tested NDIs it revealed relatively good solubility in 1,2-dichlorobenzene. This is probably due to its unique crystal packing that is discussed later in the article. 

### 2.3. Characteristic Features of NMR Spectra of the Symmetrically N-Substituted NDIs

Analysis of the ^13^C NMR spectra is intuitive and the spectral features can be unambiguously assigned to NDIs’ molecular structures. Briefly: the ^13^C NMR spectra reveal characteristic signal of the imide carbonyl group at 162 ppm, three signals of the aromatic carbons at 126–131 ppm, and a set of peaks characteristic for the alkyl chains on the NDIs’ molecules. The collection of ^13^C NMR data is provided in Appendix A. All NMR spectra are provided as a Appendix A to this article. 

The ^1^H NMR spectra (Appendix A) include a characteristic singlet from the four aromatic naphthyl protons at 8.74–8.78 ppm and a set of signals corresponding to n-alkyl chains. The signals of the alkyl chains appear in the range of 0.87–1.81 ppm and at 4.19–4.21 ppm (Appendix A). Most of the ^1^H NMR spectral features are in agreement with the literature [21,34,38,39,55,56,57,58,59,60,61]. However, the multiplicity of the signal at 4.2 ppm originating from the protons of methylene group adjacent to the imide nitrogen atoms (Figure 4) often seems misinterpreted in the literature [38,55,56,59,61]. Linear alkyl chains and the symmetry of the NDIs’ molecules intuitively suggest that coupling of the protons of the methylene group adjacent to the nitrogen atom with the two protons of the subsequent methylene group should result in a simple triplet. Hence, in the literature, the signals appearing within 4.19–4.24 ppm are typically considered triplets [38,55,59,61]. Detailed analysis of NDIC3 and NDIC5 spectra revealed, however, that the middle peak in the discussed range is flanked by two additional peaks (Figure 4a). In other NDIs, the flanking peaks partially overlap with the middle peak, which appears as middle peak broadening (Figure 4b). Thus, the signals appearing from 4.19 to 4.24 ppm should be considered a second-order spin system; that is, and, therefore, they are multiplets. The multiplet character of the signal at 4.19–4.24 ppm is similar to the AA’XX’ spin system characteristic of, e.g., aryl n-propyl sulfones [62]. On the other hand, in the case of a multiplet with broadened middle line description, assigning them as triplets is conditionally acceptable. In the case when all five lines in a multiplet are visible, referring to them as a “multiplet” or “AA’XX’ spin system” is definitely more appropriate. Moreover, magnetic non-equivalence of the two methylene protons neighboring the nitrogen atom causes increased spectral complexity of the adjacent, following the methylene group (the second from the *N*-atom, see Figure 4c).

### 2.4. Effects of Alkyl Chain Length on FTIR Spectra of Solid NDIs

Full spectra are provided in Appendix A, and the positions of the absorption bands are listed in Appendix A. The NDIs’ spectra are dominated by absorption bands characteristic of aromatic, alkyl, and imide moieties [39,63]. Selected characteristic bands are shown in Figure 5. Expectedly, the intensity of the bands corresponding to the motion of alkyl chains monotonically increases with the increasing number of carbon atoms in the alkyl chains. However, the variation in the alkyl-related bands is not only related to their intensity. Inspection of the spectra in the range of 2800 to 3000 cm^−1^ reveals an interesting transition: the position of the peak visible at υ~2940 cm^−1^ for short-alkyl chain-NDIs drops by δ_υ_ ≈ 15 cm^−1^ down to ~2925 cm^−1^ in approximately the middle range of the studied NDIs (Figure 5a). That transition can be likely attributed to a so-called gamma-gauche effect reported, for instance, for tetrathiafulvalene-based molecules [64]. Briefly, the gamma-gauche effect is related to the formation of specific packing of longer alkyl chains in regular crystal arrays. In order, however, to ultimately determine the nature of the transition observed here, a separate solid-state NMR study would be necessary [65,66]. The length of alkyl chains also exerts an influence on spectral features corresponding to the aromatic moieties of the NDI molecules. The absorption peaks related to aromatic C=C stretching (1560–1600 cm^−1^ Figure 5b) and aromatic out-of-plane bending (740–780 cm^−1^ Figure 5d) are subtly shifted towards the lower υ-numbers for NDIC3; these bands are also broader and less intense than observed in the NDIs with longer alkyl chains (Figure 5). Another interesting and, to the best of our knowledge, so-far unreported feature is related to the C-N stretching bands (1320–1350 cm^−1^, Figure 5c). The imide C-N stretching peak positions reveal a clear odd–even parity effect. Such effects related to the orientation of aromatic cores were recently reported for a homological series of double *N,N*′-bis(n-alkyl)-naphthalenediimides [33]. Here, in the case of symmetrical NDIs, the C-N stretching peaks corresponding to NDIs with odd numbered alkyl substituents are located at higher υ whereas those for the NDIs with an even number of carbon atoms in alkyl chains tend to be lower (Figure 5c). The odd–even δ_υ_ difference averages approximately 11 cm^−1^ (Figure 5). There are two exceptions from the odd–even rule: (i) NDIC6, where the band maximum is located at υ higher than excepted, and (ii) NDIC3 that shows a noticeably different spectral profile within 1320 < υ < 1350 cm^−1^ corresponding to the imide C-N stretching band (Figure 5). The variation in spectral features related to the aromatic cores of the NDIs can be explained on the basis of crystal packing discussed in the next section. 

### 2.5. Crystal Structure of NDIs

Research performed over the past 30 years provided extensive crystallographic data on various NDIs [7,19,55,56,57,67]. A survey of the related literature indicates that the research is mostly related to short-chained NDIs. Data for longer chain NDIs are less represented, so here we fill this gap. In order to complete the existing crystallographic dataset in the range from NDIC3 to NDIC11, we have determined the crystal structure of NDIC7, NDIC9, and NDIC11 (see Appendix A). Our data, together with the crystal data of NDIC3 [55], NDIC4 [56], NDIC5 [57], NDIC6 [7], NDIC8 [19], and NDIC10 [67] reported previously in literature are collected in Table 2. The results acquired by the single-crystal X-ray diffraction technique were additionally compared with the crystallographic data determined from powder patterns (Table 2, numbers in parentheses). Selected crystal data and structure refinement details of the above NDIs are extensively reported in the Appendix A. Similarly, as observed in the FTIR spectra (Figure 5), there is an interesting odd–even parity effect related to the alkyl chain length of the NDIs. The NDIs with an even number of methylene groups in the alkyl chains form typically triclinic unit cells with translational symmetry only (P$\overset{-}{1}$, see Table 2). The crystals of NDIs substituted with odd-numbered alkyl chains reveal higher symmetry—they form typically P2_1_/c monoclinic cells (Table 2). The odd–even parity effect also manifests itself in the π-π stacking distance (d_π-π_, see Appendix A). The odd-numbered NDIs generally reveal slightly smaller d_π-π_ values (3.2 Å on average) then the even-numbered ones (3.3 Å on average). In terms of crystal structure the NDIC3 is the most different NDI in the whole NDIC3–NDIC11 range. The NDIC3 has an orthorhombic (Pbca) unit cell and it was the only NDI in the study showing non-parallel stacking of the aromatic cores. That non-parallel arrangement explains the difference observed in the C-N stretching band in the FTIR spectrum of the NDIC3 (Figure 5). The parameters of the unit cells and their formulae were used to Miller-index the powder patterns (see Appendix A). Room-temperature crystal structures of NDI powders (and films) were found identical with the crystal structures of the single crystals. 

### 2.6. Crystal Morphologies of NDIs

One of the valid issues in modern technologies including fields of, among others, organic electronics, photovoltaics, or sensor systems is the fabrication of thin films. On a technological scale, the films are often fabricated by sublimation [21,68]. Nevertheless, however, a challenge is to fabricate films via scalable solution-based technologies like inkjet or screen printing or spray- or meniscus-guided coating [19]. It is, therefore, important to know the solubility—as provided here in Section 2.2—as well as to know the crystal morphologies resulting from isotropic crystallization—i.e., crystallization unaffected by any kind of external force. Here in Figure 6 we provide the polarizing optical morphologies of NDI films crystallized from chloroform at 25 °C on glass substrates. Generally, the NDIs in the study form two kind of morphologies: spherulitic/acicular for NDIs with shorter aliphatic chains and lamellar/feather-like morphologies for NDIs with longer aliphatic chains. NDIC3 and NDIC4 showed radiating domains with spherulitic morphology, whereas, in the case of NDIC4 on the images of the thicker part of the layer, needle and acicular structures could be also distinguished. Crystallization of NDIs C5, C6, and C7 resulted in fan-type sheets with fine lamellar textures. The NDIs C9, C10, and C11 show lamellar feather-like structures. An inspection of the micrographs in Figure 6 generally indicates that NDIs tend to dewet the glass leading to issues with film continuity. It seems that NDIC5, NDIC6, and NDIC7 formed the most continuous films after crystallization from chloroform. 

### 2.7. Thermal Properties of Solid NDIs

NDIs are relatively thermally stable molecules. TGA measurements indicated that non-oxidative thermal decomposition of NDIs begins above 300 °C—i.e., beyond the melting point. Onsets of thermal decomposition (T_−3_, T_−5_, and T_−10_, see Figure 7 and Appendix A) of the NDIs were found increasing along with the increasing number of carbon atoms in the alkyl chains. Interestingly, the onset of thermal decompositions of NDIC7 was negatively deviated from a generally linear trend. That negative deviation was, to some extent, qualitatively similar to a deviation of NDIC7’s solubility in heptane (Figure 3a). The TGA profiles of NDIs (see Appendix A) suggest that the thermal decomposition of NDIs is a single-step process. 

Further investigations of the NDIs’ thermal properties were performed with differential scanning calorimetry (DSC). Some DSC studies of NDIs were already reported in literature [56,67,69]. In an attempt to find the general pattern of the phase transitions, the whole range of the NDIs in this work was analyzed in a unified way. The recorded raw experimental DSC curves are shown in Appendix A. Phase transition points and transition enthalpies are collected in Appendix A where they are compared with the available literature data. In order to provide a clear overview of melting enthalpies and phase transition points for all NDIs, the data are graphically presented in Figure 8. The transition enthalpies were calculated from integrals over corresponding endo- or exotherms. Then, the calculated enthalpies were charted against temperature (Figure 8). The transition enthalpies in Figure 8 are marked in the shades of red or blue to distinguish between endothermic (red) or exothermic (blue) phase transitions. As indicated in the color scale bar, darker shades indicate higher enthalpy (Figure 8). Heating in the range from −20 °C to 250 °C causes the NDIs to reversibly transform two or three times before the melting (isotropization). Most notably, the isotropization temperature of NDIs decreases monotonically with increasing alkyl chain lengths (Figure 8). Unlike double *N*,*N*′-bis(n-alkyl)-naphthalenediimides, and despite a regular odd–even pattern of crystal unit cells (see Appendix A), NDIs do not show any kind of odd–even parity effect on melting or crystallization. However, and quite interestingly, the transformation points preceding the isotropization form two distinct trends. In the range of NDIC3–NDIC7, a trend roughly following the line from 100 °C to 150 °C is observed. In the range of NDIC7–NDIC11, another set of transitions following the line connecting −10 °C (for NDIC7) and 100 °C (for NDIC11) can be discerned. NDIC7 and NDIC8 revealed the most complex phase behavior. Based on the melting enthalpies and reversibility, the phase transitions preceding isotropization were identified as mostly crystal–crystal polymorphic transformations (Appendix A). In some cases, such as NDIC4, the phase transitions preceding isotropization can be attributed to the crystal to liquid crystal transition [67]. 

### 2.8. Field-Effect Electron Mobility

Solution-processed, linear alkyl-substituted NDIs exhibit relatively high field-effect electron mobilities (μ_e_), reportedly reaching almost 0.2 cm^2^V^−1^s^−1^ [17,42,50,70,71]. The μ_e_ for NDIC8 (studied in this work) can reach 0.16 cm^2^V^−1^s^−1,^ whereas longer-chain NDIs tend to reveal lower mobilities: μ_e_ for NDIC12 and NDIC18 (not included in the NDI range in this work) were reported to be 0.01 and 0.005 cm^2^V^−1^s^−1^, respectively [42,72]. Despite a proven potential in the field of electronic and energy materials as n-type semiconductors [54], a survey of the literature reveals that a complete comparative study of their electronic properties is still lacking. Analysis of the scattered published data indicates, for instance, that the effect of the alkyl chain length cannot be unequivocally identified based on previous studies mainly because of different methodologies used in different works [51,70]. Hence, in our work, we performed unified tests of electron field mobility using organic field-effect transistors (OFETs) as diagnostic devices. For the sake of comparison, we have used completely unified and fully reproducible experimental procedures: the NDI films were deposited on commercially available OFET substrates with prepatterned gold electrodes. In order to avoid uncontrolled phase transitions and dewetting of NDI films (see Section 2.7) the OFETs were neither thermally nor solvent-vapor annealed. The solvent (chloroform) was removed by placing the OFET in a mild vacuum. Obviously, because of unequilibrated crystal morphology and a mismatch between gold work function (−5.1 eV) and HOMO/LUMO levels of NDIs (approx. −7.2 eV/−3.6 eV, respectively), our methodology did not allow reaching the highest possible μ_e_ or the lowest possible threshold voltage (V_th_) numbers [20,70,71,73]. On the other hand, however, our approach enabled a direct comparison between all NDIs in the studied range. The data collected in Table 3 suggest that the V_th_ ranges from 40 to 55 V and reveal no clear trend. The recorded V_th_ can be considered typical for molecular, NDI-like materials [70,71]. Unlike the V_th_, μ_e_ numbers reveal the optimum range of the alkyl chain length. Comparison of the data in Table 3 indicates that the range of studied NDIs’ μ_e_ number reaches a maximum for NDIC6, NDIC7, and NDIC8. For these three NDIs, the μ_e_ is in the range of 10^−4^ cm^2^V^−1^s^−1^ whereas, for the NDIs substituted with chains shorter or longer than in the abovementioned range, the μ_e_ tends to be lower by one order of magnitude (Table 3). 

## 3. Materials and Methods

### 3.1. Materials 

Imidazole >98% and 1,4,5,8-naphthalenetetracarboxylic dianhydride (>95%) were purchased from TCI Europe N.V. Toluene, hydrochloric acid, 96% ethanol, and methanol were obtained from Avantor Performance Materials Poland. Chloroform-D (CDCl_3_), n-aliphatic amines, n-heptane, toluene (CHROMASOLV^®^ (Honeywell, Charlotte, NC, USA), for HPLC, 99.9%), 1,2-dichlorobenzene (for HPLC, 99%), and chloroform (CHROMASOLV^®^Plus, (Honeywell, Charlotte, NC, USA), for HPLC, >99.9%) were supplied by Sigma-Aldrich, Darmstadt, Germany. All reagents and solvents were used as received. The OFETs were prepared using silicon substrates with 230 nm silicon dioxide gate dielectric with prepatterned Au/ITO (20 nm/10 nm) electrodes (Fraunhofer Gen. 4 OFET substrates (Fraunhofer IPMS, Dresden, Germany)). The OFET channel width (W) and the length were 10 mm and 10 μm, respectively.

### 3.2. Synthesis

In the study, we adopted the following synthetic protocol: a mixture of 1,4,5,8-naphthalenetetracarboxylic dianhydride (4.0 g, 15 mmol), imidazole (8.2 g, 120 mmol), and appropriate amine (45 mmol in the case of *n*-pentylamine to *n*-undecylamine and 120 mmol of *n*-butylamine and *n*-propylamine) in toluene (75 mL) were stirred and heated under reflux at 150–155 °C (oil bath temperature) for about 12 h. The reaction mixture was cooled to room temperature and evaporated to dryness. The obtained residue was suspended in 96% ethanol (~150 mL) and gradually added to vigorously stirred, cold 2N hydrochloric acid (~300 mL). The resulting suspension was stirred for about 1 h and filtered directly through a Soxhlet thimble. The thimble, together with the crude product, was placed in a Soxhlet extractor body (200 mL, Quickfit^®^ (DWK Life Sciences Ltd., Stoke-on-Trent, UK)) equipped with a 500 mL round bottom flask and reflux condensers open to the air. In order to remove any colored impurities and starting materials, the crude product was extracted with methanol (300 mL) for 4–8 h. The heating was controlled to maintain the extraction rate at the level of approx. one drop of methanol per second. The extraction was continued 1 h beyond the point at which the washings became colorless (or almost colorless). Afterwards, the extraction solvent was changed to chloroform (300 mL) and the extraction was continued until all material soluble in chloroform was transferred to the flask. The chloroform solution of the NDI was then evaporated to dryness, and the residue was treated with methanol (50 mL), filtered, and dried in air. In the case of NDIC3 and NDIC4, additional purification via short-column chromatography and sublimation was required. Hence, NDIC3 and NDIC4 were additionally dissolved in a small amount of CH_2_Cl_2_, passed through a pad of silica gel (20 g, 70–230 mesh, Sigma-Aldrich), and finally washed with approx. 150 mL of CH_2_Cl_2_. Evaporation and sublimation (200 °C, 1.2∙10^−2^ mmHg) resulted in pure NDIC3 and NDIC4.

The chemical identity and purity of NDIs were confirmed by ^1^H and ^13^C NMR spectroscopy. The spectra were in agreement with the literature data (NDIC3 [34,55]; NDIC4 [56,57]; NDIC5 [2,34,38]; NDIC6 [58,59]; NDIC7 [34]; NDIC8 [39,56]; NDIC9 [21,34]; NDIC10 [35,60]; and NDIC11 [58,61]). Copies of all the NMR spectra are provided in the Appendix A. The spectral data are listed below and provided in Appendix A.

*N*,*N*′-di(n-propyl)naphthalene-1,4,5,8-tetracarboxylic diimide (**NDIC3**): yield = 35%; ^1^H NMR (600 MHz, CDCl_3_) δ: 1.03 (t, 6H, *J* = 7.4 Hz, CH_3_), 1.78 (sext, 4H, *J* = 7.5 Hz, CH_3_CH_2_), 4.17 (m (AA’XX’ spin system), 4H, CH_2_N), 8.75 (s, 4H, ArH); ^13^C NMR (151 MHz, CDCl_3_) δ: 11.49, 21.38, 42.42, 126.61, 126.67, 130.92, 162.83; FTIR: 3076, 3061, 3043, 2953, 2920, 2850, 1699 (C=O), 1653 (C=O), 1581 (C=C), 1454, 1373, 1337 (C-N), 1277, 1257, 1242, 1200, 1155, 1080, 1026, 964, 890, 806, 770, 720, 608, 565, 440, 407 cm^−1^.

*N*,*N*′-di(n-butyl)naphthalene-1,4,5,8-tetracarboxylic diimide (**NDIC4**): yield = 51%; ^1^H NMR (600 MHz, CDCl_3_) δ: 1.02 (t, 6H, *J* = 7.4 Hz, CH_3_), 1.48 (sext, 4H, *J* = 7.5 Hz, CH_3_CH_2_), 1.76 (m, 4H, CH_2_CH_2_N), 4.23 (m (AA’XX’ spin system), 4H, CH_2_N), 8.78 (s, 4H, ArH); ^13^C NMR (151 MHz, CDCl_3_) δ: 13.81, 20.35, 30.17, 40.76, 126.64, 126.69, 130.92, 162.84; FTIR: 3076, 3063, 3040, 2960, 2939, 2873, 2858, 1699 (C=O), 1654 (C=O), 1578 (C=C), 1450, 1373, 1329 (C-N), 1301, 1286, 1246, 1219, 1191, 1147, 1076, 982, 970, 890, 856, 768, 744, 717, 640, 594, 538, 488, 415 cm^−1^.

*N*,*N*′-di(n-pentyl)naphthalene-1,4,5,8-tetracarboxylic diimide (**NDIC5**): yield = 88%; ^1^H NMR (600 MHz, CDCl_3_) δ: 0.94 (t, 6H, *J* = 7.1 Hz, CH_3_), 1.38–1.48 (m, 8H, CH_3_(CH_2_)_2_), 1.77 (m, 4H, CH_2_CH_2_N), 4.21 (m (AA’XX’ spin system), 4H, CH_2_N), 8.76 (s, 4H, ArH); ^13^C NMR (151 MHz, CDCl_3_) δ: 13.96, 22.38, 27.71, 29.16, 40.91, 126.51 (two overlapped signals), 130.82, 162.67; FTIR: 3074, 3063, 3041, 2958, 2935, 2874, 2858, 1701 (C=O), 1650 (C=O), 1580 (C=C), 1454, 1375, 1340 (C-N), 1267, 1244, 1180, 1148, 1084, 1063, 1030, 1000, 949, 894, 864, 844, 808, 770, 735, 710, 617, 567, 438, 405 cm^−1^.

*N*,*N*′-di(n-hexyl)naphthalene-1,4,5,8-tetracarboxylic diimide (**NDIC6**): yield = 82%; ^1^H NMR (600 MHz, CDCl_3_) δ: 0.92 (t, 6H, *J* = 7.2 Hz, CH_3_), 1.33–1.41 (m, 8H, CH_3_(CH_2_)_2_), 1.46 (m, 4H, CH_2_(CH_2_)_2_N), 1.77 (m, 4H, CH_2_CH_2_N), 4.21 (m (AA’XX’ spin system), 4H, CH_2_N), 8.78 (s, 4H, ArH); ^13^C NMR (151 MHz, CDCl_3_) δ: 14.04, 22.54, 26.74, 28.02, 31.49, 40.98, 126.60, 126.64, 130.87, 162.78; FTIR: 3076, 3061, 3041, 2964, 2916, 2852, 1697 (C=O), 1652 (C=O), 1580 (C=C), 1454, 1375, 1338 (C-N), 1242, 1175, 1153, 1084, 1051, 1022, 1000, 960, 894, 808, 770, 710, 619, 567, 438, 417 cm^−1^.

*N*,*N*′-di(n-heptyl)naphthalene-1,4,5,8-tetracarboxylic diimide (**NDIC7**): yield = 75%; ^1^H NMR (600 MHz, CDCl_3_) δ: 0.90 (t, 6H, *J* = 7.0 Hz, CH_3_), 1.27–1.36 (m, 8H, CH_3_(CH_2_)_2_), 1.39 (m, 4H, CH_2_(CH_2_)_3_N), 1.45 (m, 4H, CH_2_(CH_2_)_2_N), 1.76 (m, 4H, CH_2_CH_2_N), 4.21 (m (AA’XX’ spin system), 4H, CH_2_N), 8.76 (s, 4H, ArH); ^13^C NMR (151 MHz, CDCl_3_) δ: 14.07, 22.59, 27.04, 28.08, 28.98, 31.73, 40.98, 126.57, 126.60, 130.87, 162.74; FTIR: 3078, 3061, 3041, 2951, 2924, 2868, 2852, 1699 (C=O), 1653 (C=O), 1580 (C=C), 1454, 1375, 1338 (C-N), 1265, 1244, 1175, 1153, 1086, 1045, 966, 808, 770, 723, 615, 567, 438, 424 cm^−1^.

*N*,*N*′-di(n-octyl)naphthalene-1,4,5,8-tetracarboxylic diimide (**NDIC8**): yield = 85%; ^1^H NMR (600 MHz, CDCl_3_) δ: 0.90 (t, 6H, *J* = 7.1 Hz, CH_3_), 1.25–1.35 (m, 12H, CH_3_(CH_2_)_3_), 1.39 (m, 4H, CH_2_(CH_2_)_3_N), 1.45 (m, 4H, CH_2_(CH_2_)_2_N), 1.76 (m, 4H, CH_2_CH_2_N), 4.21 (m (AA’XX’ spin system), 4H, CH_2_N), 8.77 (s, 4H, ArH); ^13^C NMR (151 MHz, CDCl_3_) δ: 14.09, 22.64, 27.10, 28.09, 29.19, 29.29, 31.80, 41.00, 126.62, 126.65, 130.89, 162.79; FTIR: 3088, 3070, 3043, 2955, 2924, 2868, 2848, 1703 (C=O), 1655 (C=O), 1580 (C=C), 1452, 1374, 1333 (C-N), 1251, 1238, 1223, 1180, 1154, 1084, 1059, 1016, 971, 889, 768, 754, 723, 607, 565, 438, 420 cm^−1^.

*N*,*N*′-di(n-nonyl)naphthalene-1,4,5,8-tetracarboxylic diimide (**NDIC9**): yield = 78%; ^1^H NMR (600 MHz, CDCl_3_) δ: 0.87 (t, 6H, *J* = 7.2 Hz, CH_3_), 1.22–1.35 (m, 16H, CH_3_(CH_2_)_4_), 1.36 (m, 4H, CH_2_(CH_2_)_3_N), 1.43 (m, 4H, CH_2_(CH_2_)_2_N), 1.74 (m, 4H, CH_2_CH_2_N), 4.19 (m (AA’XX’ spin system), 4H, CH_2_N), 8.75 (s, 4H, ArH); ^13^C NMR (151 MHz, CDCl_3_) δ: 14.11, 22.66, 27.09, 28.09, 29.26, 29.33, 29.49, 31.85, 41.00, 126.61, 126.65, 130.90, 162.79; FTIR: 3076, 3061, 3041, 2951, 2918, 2850, 1699 (C=O), 1652 (C=O), 1581 (C=C), 1454, 1373, 1337 (C-N), 1270, 1246, 1221, 1178, 1155, 1080, 1020, 977, 964, 893, 806, 768, 729, 719, 608, 565, 440, 419 cm^−1^.

*N*,*N*′-di(n-decyl)naphthalene-1,4,5,8-tetracarboxylic diimide (**NDIC10**): yield = 79%; ^1^H NMR (600 MHz, CDCl_3_) δ: 0.87 (t, 6H, *J* = 7.2 Hz, CH_3_), 1.20–1.33 (m, 20H, CH_3_(CH_2_)_5_), 1.36 (quint, 4H, *J* = 7.2 Hz, CH_2_(CH_2_)_3_N), 1.43 (m, 4H, CH_2_(CH_2_)_2_N), 1.74 (m, 4H, CH_2_CH_2_N), 4.19 (m (AA’XX’ spin system), 4H, CH_2_N), 8.75 (s, 4H, ArH); ^13^C NMR (151 MHz, CDCl_3_) δ: 14.11, 22.68, 27.09, 28.07, 29.30, 29.33, 29.53, 29.55, 31.88, 40.98, 126.56, 126.58, 130.86, 162.72; FTIR: 3087, 3061, 3043, 2955, 2922, 2847, 1703 (C=O), 1655 (C=O), 1581 (C=C), 1454, 1373, 1331 (C-N), 1263, 1246, 1217, 1174, 1153, 1088, 1074, 1026, 978, 964, 891, 806, 768, 723, 608, 565, 438, 409 cm^−1^.

*N*,*N*′-di(n-undecyl)naphthalene-1,4,5,8-tetracarboxylic diimide (**NDIC11**): yield = 68%; ^1^H NMR (600 MHz, CDCl_3_) δ: 0.87 (t, 6H, *J* = 7.1 Hz, CH_3_), 1.20–1.32 (m, 24H, CH_3_(CH_2_)_6_), 1.36 (quint, 4H, *J* = 7.4 Hz, CH_2_(CH_2_)_3_N), 1.43 (m, 4H, CH_2_(CH_2_)_2_N), 1.74 (m, 4H, CH_2_CH_2_N), 4.19 (m (AA’XX’ spin system), 4H, CH_2_N), 8.75 (s, 4H, ArH); ^13^C NMR (151 MHz, CDCl_3_) δ: 14.12, 22.70, 27.10, 28.10, 29.33 (two overlapped signals), 29.53, 29.60 (two overlapped signals), 31.92, 41.01, 126.65, 126.69, 130.92, 162.83; FTIR: 3076, 3061, 3043, 2953, 2918, 2848, 1699 (C=O), 1653 (C=O), 1581 (C=C), 1454, 1375, 1335 (C-N), 1278, 1257, 1242, 1200, 1155, 1080, 1030, 964, 891, 808, 768, 729, 609, 567, 438, 409 cm^−1^.

### 3.3. Preparation of Isotropic Samples

The isotropic samples for infrared spectroscopy (ATR FT-IR), differential scanning calorimetry (DSC), thermogravimetric analysis (TGA), and powder X-ray diffraction (PXRD) measurements were crystallized from chloroform on Petri dishes to form micron-thick films. The films were then cut into fine pieces (0.1–0.5 mm). Samples for the polarized optical microscopy (POM) were prepared by drop casting of the solutions followed by slow crystallization on microscopic glass slides under chloroform vapor atmosphere. Single crystals of NDIC7, NDIC9, and NDIC11 suitable for X-ray diffraction were grown by slow evaporation of a solution in chloroform.

### 3.4. Infrared Spectroscopy

The infrared spectra (FTIR) were recorded for isotropic samples in the diamond ATR cell. In the study, we used the Bruker Vertex 70v Fourier transform IR spectrometer (Billerica, MA, USA). The spectra were collected as 64 scans at room temperature over the range of 4000 to 400 cm^−1^ with 2 cm^−1^ resolution. The measurements, the instrument control, and initial data processing were performed using OPUS software (v. 7.0 Bruker Optics, Ettlingen, Germany). The spectra were processed and visualized using OriginPro 2019.

### 3.5. UV-Vis Spectroscopy and Solubility Studies

The UV-Vis absorption spectra of the NDIs were measured in the chloroform (0.10 mM). The spectra were collected using a Perkin Elmer Lambda 20 spectrophotometer with a spectral resolution of 1 nm (Waltham, MA, USA). The NDIs were dissolved in chloroform and measured in quartz cuvettes with 10 mm or 1 mm path length at 20 °C. Data processing was performed using UV WinLab 2.70 software. For the purpose of data presentation, the UV-Vis spectra were exported to OriginPro 2019 as ASCII files. In order to test the solubility, 100.0 mg of each of the NDIs was dissolved in 1.0 mL of chloroform. For NDIs that dissolved completely, their solubility was considered as >100 mg/mL [49]. In case of incomplete dissolution, the solubility was determined quantitatively. The quantitative solubility determination was performed for selected NDIs in chloroform and all NDIs in n-heptane, toluene, and 1,2-dichlorobenzene. In order to determine the solubility, the NDIs were volumetrically dissolved in n-heptane, toluene, 1,2-dichlorobenzene, or chloroform to obtain stock solutions with concentrations of 0.040 mg/mL, 0.400 mg/mL, or 0.400 mg/mL, or 3.000 mg/mL, respectively. Then, the stock solutions were diluted to obtain five standard solutions. Concentrations of standard solutions were: 0.0200 mg/mL, 0.0133 mg/mL, 0.0100 mg/mL, 0.0080 mg/mL, and 0.0067 mg/mL in n-heptane; 0.2000 mg/mL, 0.1330 mg/mL, 0.1000 mg/mL, 0.080 mg/mL, and 0.0667 mg/mL in toluene and 1,2-dichlorobenzene; or 1.500 mg/mL, 1.000 mg/mL, 0.750 mg/mL, 0.600 mg/mL, and 0.500 mg/mL in chloroform. For each standard solution, a UV-Vis spectrum in the λ-range from 230 nm to 400 nm was measured. The absorbances at 376 nm (for n-heptane solutions), 281 nm (toluene), 363 nm (1,2-dichlorobenzene), or 265 nm (chloroform) were calibrated against the concentration. Saturated solutions were prepared by mixing NDIs (in amounts exceeding solubility) with 1 mL of each of the above solvents. The mixtures were stirred for 24 h at 25 °C. Afterwards, the saturated solutions were filtered through a 0.2 μm PTFE syringe filter and their absorbance at the corresponding wavelength was measured. The concentrations of the saturated solutions were determined from the absorbance using the calibration plots obtained for the corresponding standard solution series. The validity of the Lambert-Beer law was confirmed by measuring the absorbance of almost saturated solutions and checking whether they follow the calibration line. No deviations from the Lambert-Beer law were observed in the whole analyzed range.

### 3.6. NMR Spectroscopy

The ^1^H and ^13^C NMR spectra were measured on the 600 MHz Bruker Avance spectrometer in CDCl_3_ (~15 mg/mL in the case of NDIC11 and ~30–40 mg/mL in the case of other NDIs) as the solvent. Chemical shifts (δ, ppm) were determined relative to CDCl_3_ signals (7.26 ppm for ^1^H NMR and 77.16 ppm for ^13^C NMR spectra). 

### 3.7. Thermogravimetric Analysis and Differential Scanning Calorimetry

Thermogravimetric analysis (TGA) was performed using the Mettler Toledo (Columbus, OH, USA) TGA/DSC1 system, calibrated with indium and zinc as standards. The tests were carried out in a temperature range of 25–600 °C with a heating rate of 10 °C/min under an argon atmosphere at a flowrate of 60 mL/min. The analyzed samples (3–11 mg) were placed in the ceramic crucibles (70 μL). Differential scanning calorimetry (DSC) measurements were performed using the Mettler Toledo DSC1 coupled with the Huber TC 100 intracooler. The instrument was calibrated using indium standard (T_m_ = 156.6 °C, ΔH_m_ = 28.45 J/g). NDI samples (2–12 mg) were sealed in 40 μL aluminum pans and heated under a constant nitrogen purge (60 mL/min) from −60 °C to the temperature above the isotropization points of NDIs. After the isotropization, the samples were thermally equilibrated for 2 min and then cooled down to −60 °C. The samples were heated and cooled at a rate of 10 °C/min. Data processing was performed using the generic Mettler Toledo STAR^e^ software and visualized using OriginPro 2019. 

### 3.8. Polarized Optical Microscopy

All micrographs were taken with crossed polarizers using the Olympus BX60 (Tokyo, Japan) equipped with a ×10 magnification lens and a Panasonic DMC-G2 camera (Kadoma, Japan).

### 3.9. X-ray Diffraction Measurements

The single-crystal diffraction (SCD) data were collected on a New Xcalibur EosS2 diffractometer (Rigaku, Tokyo, Japan). The patterns were recorded using monochromatic MoK_α_ radiation (λ = 0.71073 Å) for NDIC11 and NDIC7 and AgK_α_ radiation (λ = 0.56087 Å) for NDIC9. The data were corrected for Lorentz polarization as well as for absorption effects [74]. The structures were solved by direct methods using the OLEX program with a plugin using the SHELXT program [75], which used an intrinsic phasing solution method, and refined with the SHELXL using least squares minimization [76]. All non-hydrogen atoms were refined anisotropically, while all hydrogen atoms were placed in idealized positions and refined as the ‘riding model’. The isotropic NDIs samples were packed in 1 mm Mark capillaries (Capillary Tube Supplies Ltd., London, UK). The XRD measurements were performed using the Ganesha 300 XL+ (SAXSLAB ApS, Kopenhagen, Denmark). The instrument was equipped with a microfocus copper source, generic parabolic multilayer mirror optics, and two x/y JJ X-ray scatterless slits providing parallel monochromatic (λ = 1.5482 Å), point-collimated beams. The diffraction patterns were recorded on the Pilatus 300 k detector (Dectris AG, Baden-Daettwil, Switzerland) with 1800 s exposure. The camera length and, therefore, the 2θ-range were calibrated using silver behenate. Conversion to one-dimensional diffraction profiles (powder patterns) was performed by integration of 2D patterns over 180° azimuthal angle (χ) using the Datasqueeze 3.0.9 program. Intensity normalization and background correction were performed numerically using OriginPro 2019. 

### 3.10. Determination of Field-Effect Electron Mobility

Field-effect electron mobility was characterized using thin-film organic field-effect transistors (OFETs) as diagnostic devices. OFETs with bottom gate, bottom contacts configuration were used. The NDI active films were spin-coated from 5 mg/mL chloroform solutions at 2500 rpm for 30 s. In order to avoid phase transitions or sublimation, the OFETs were not thermally annealed or vacuum dried. All operations and measurements related to OFETs were performed under nitrogen atmosphere in the glove box system. The OFET measurements were performed with the Keithley 2614B dual-source meter (Tektronix Inc., Beaverton, OR, USA). Output characteristics were measured in the range from −10 to 80 V with 5 V gate-source step and 2 V drain-source step. The electron mobility was extracted from transfer characteristics in the saturation regime according to Equation (1).
(1)$$\mu_{sat}=\frac{2L}{WC_{i}}{\left(\frac{\partial \sqrt{I_{DS}}}{\partial V_{G}}\right)}^{2}$$
where I_DS_ is the drain current, W is the channel width, L is the channel length (see Section 3.1), C_i_ is the capacitance of gate dielectric, and V_g_ is the gate voltage.

## 4. Conclusions

Herein, we present a study on a homological series of NDIs *N*-substituted with linear aliphatic chains with 3 to 11 carbon atoms. The NDIs were analyzed spectroscopically with UV-VIS, NMR, and FTIR. The research was supplemented with studies on crystal structure and morphology performed by, respectively, X-ray diffraction (both powder and single-crystal) and optical microscopy. Our systematic approach permitted unraveling interesting trends in the whole homological series as well as some peculiar features of specific NDIs. We started our studies with a determination of solubility in solvents commonly used in the fabrication of organic electronics films. It turned out that NDIs in the range of NDIC5–NDIC8 revealed the best of all solubility in chloroform, dichlorobenzene, or toluene. Extending the length of alkyl substituents beyond 8 carbon atoms caused a 2- or 3-fold decrease in the solubility in each of the solvents. Generally, the NDIs with shorter alkyl substituents also reveal solubility significantly lower than that observed in the medium-length alkyl substituents. Observations on solubility already at this stage make the range of NDIC5–NDIC8 distinct from the viewpoint of practical applications. The effect of alkyl chain length on the solubility of NDIs in n-heptane is different. Extending the alkyl substituent chain length resulted in a gradual increase in the solubility in n-heptane from 0.005 mg/mL (NDIC3) to 0.600 mg/mL (NDIC11). However, the solubility of short- and long-alkyl-chained NDIs in n-heptane follow slightly different trends. The solubility of NDIs with substituents up to 6 carbon atoms increases faster with the number of carbon atoms. Starting from the the n-heptyl substituted NDIC7, we observed a different, linear trend in solubility with increasing length of the alkyl substituents. Our solution NMR studies indicated that the multiplicity of the methylene protons adjacent to the nitrogen atom is a non-first-order AA’XX’ system and not a simple triplet, as is typically reported in literature. This observation suggests more complex than intuitively perceptible interactions of alkyl chains with imide moieties of the NDIs. The FTIR analysis of the solid NDIs revealed interesting spectral features related to the aromatic cores of the NDIs. The imide C-N stretching peak positions revealed a clear odd–even parity effect related to the length and number of methylene groups in the alkyl substituents. For the NDIs with odd numbered alkyl substituents, the C-N stretching peaks were found at wavenumbers distinctly higher than the peaks of the even-numbered NDIs. Further FTIR analysis revealed that NDIs can be divided into two distinct subgroups: the short- and the long-chained. The NDIs in the middle of the analyzed range (i.e., NDIC5–NDIC7) reveal some additional characteristic spectral features. Crystal packing of NDIs revealed a clear odd–even parity effect, thus explaining the spectroscopy findings. The NDIs with an even number of methylene groups in the alkyl substituents revealed low-symmetry (P$\overset{-}{1}$) triclinic packing, whereas those with an odd number of carbon atoms were generally monoclinic with P2_1_/c symmetry. The odd–even parity effect manifests itself in an overlapping of the NDIs’ aromatic cores and the π-π stacking distance (d_π-π_). The odd-numbered NDIs generally reveal slightly smaller d_π-π_ values then the even-numbered ones. Solid NDIs revealed relatively high thermal stability that was found increasing from 320 °C to 420 °C with increasing substituent chain length. Interestingly, here again, some deviation from a linear trend was observed in the middle range of the alkyl chain length. The NDIs revealed remarkable patterns of thermally induced phase transitions. All the NDIs show a series of solid-state transitions prior to the isotropization. We observed two distinct patterns (kinds) of these pre-melting transitions for short- and long-chained NDIs. The NDIC7, exactly in the middle of the studied range, revealed transitions of both kinds. The NDIs from the middle of the analyzed range also stand out in terms of electronic properties. The field-effect electron mobilities determined for the NDIC6–NDIC8 range in normalized OFETs were 10- to 30-fold higher than for NDIs with shorter or longer alkyl substituents. All these observations may suggest that balanced aromatic and alkyl interactions are the key to achieve the best solution processability and electronic performance. 

## Figures and Tables

**Figure 1 molecules-28-02940-f001:**
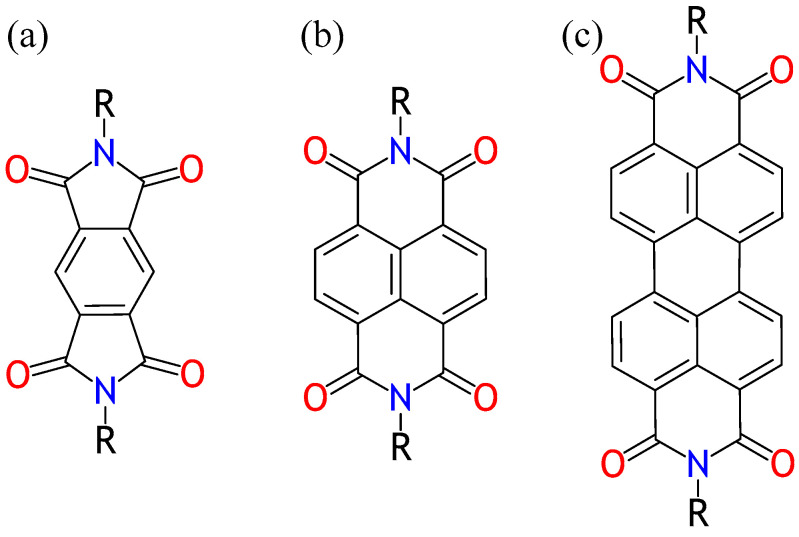
Chemical structures of *N*,*N*′-dialkylated pyromellitic diimides (PyDI, (**a**)); *N*,*N*′-dialkylated naphthalene diimides (NDI, (**b**)); and *N*,*N*′-dialkylated perylene diimides (PDI, (**c**)). R-substituents are n-alkyl chains. For the NDIs described in this paper, the R may include from 3 through 11 carbon atoms in a linear alkyl chain.

**Figure 2 molecules-28-02940-f002:**
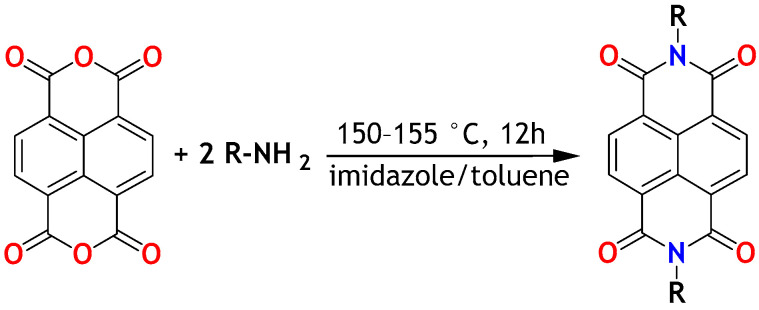
Synthesis of NDIs. R = n-propyl (NDIC3); R = n-butyl (NDIC4); R = n-pentyl (NDIC5); R = n-hexyl (NDIC6); R = n-heptyl (NDIC7); R = n-octyl (NDIC8); R = n-nonyl (NDIC9); R = n-decyl (NDIC10); and R = n-undecyl (NDIC11).

**Figure 3 molecules-28-02940-f003:**
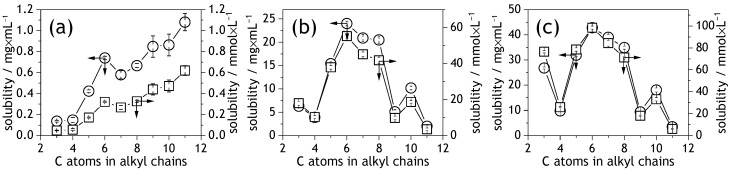
Solubility of NDIs in n-heptane (**a**), toluene (**b**), and 1,2-dichlorobenzene (**c**). Circles correspond to solubility in mg/mL (left axis). Square symbols indicate solubility in mmol/L (right axis).

**Figure 4 molecules-28-02940-f004:**
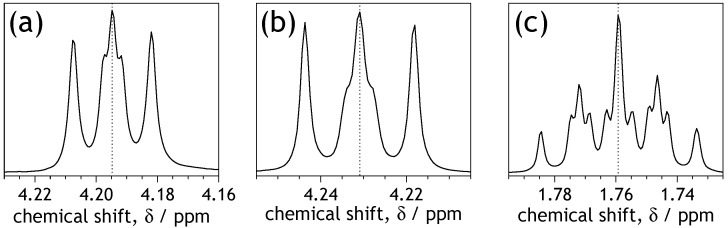
Examples of ^1^H NMR signal profiles of: (**a**) NCH_2_ protons of NDIC3 with clearly indicated five lines of multiplet; (**b**) NCH_2_ protons of NDIC4 with broadening of a middle line of multiplet (note the shift at the δ-scale); and (**c**) β–methylene group protons (NCH_2_CH_2_CH_2_) of NDIC4.

**Figure 5 molecules-28-02940-f005:**
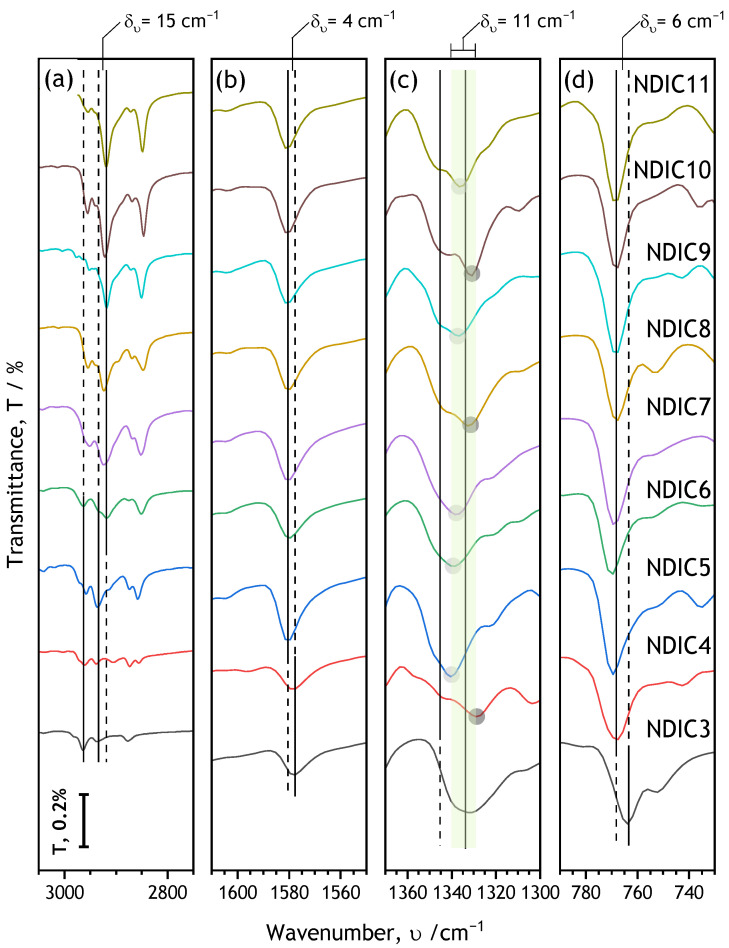
Characteristic regions of FTIR spectra (Full spectra are shown in Appendix A; see also Appendix A). For clarity, the spectra are shifted along the transmittance scale. The common transmittance scale bar is in the lower left corner. For the range of NDIs where the absorption peaks are observed at certain constant wavenumber (υ) the peak positions are marked with vertical solid lines. Dashed lines serve as guides to the eye. See the text for assignments of peaks to vibrations.

**Figure 6 molecules-28-02940-f006:**
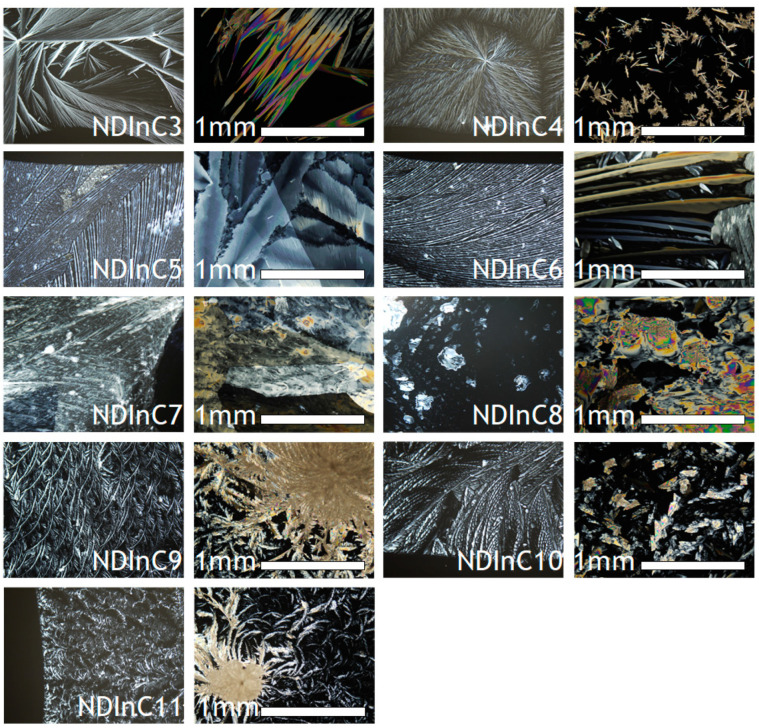
Morphology of the studied NDIs deposited at 25 °C on glass plates as revealed by polarized optical microscopy.

**Figure 7 molecules-28-02940-f007:**
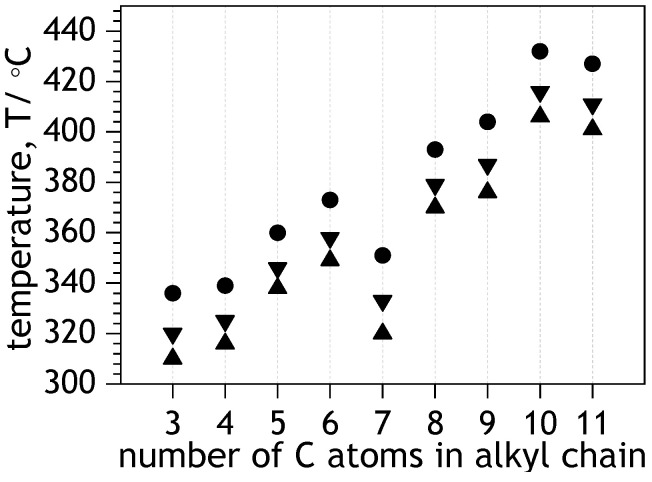
Thermal stability of the studied NDIs in the function of carbon atoms in alkyl chain. Symbols ▲, ▼ and ● indicate 3%, 5%, and 10% weight loss: T_−3_, T_−5_, and T_−10_, respectively.

**Figure 8 molecules-28-02940-f008:**
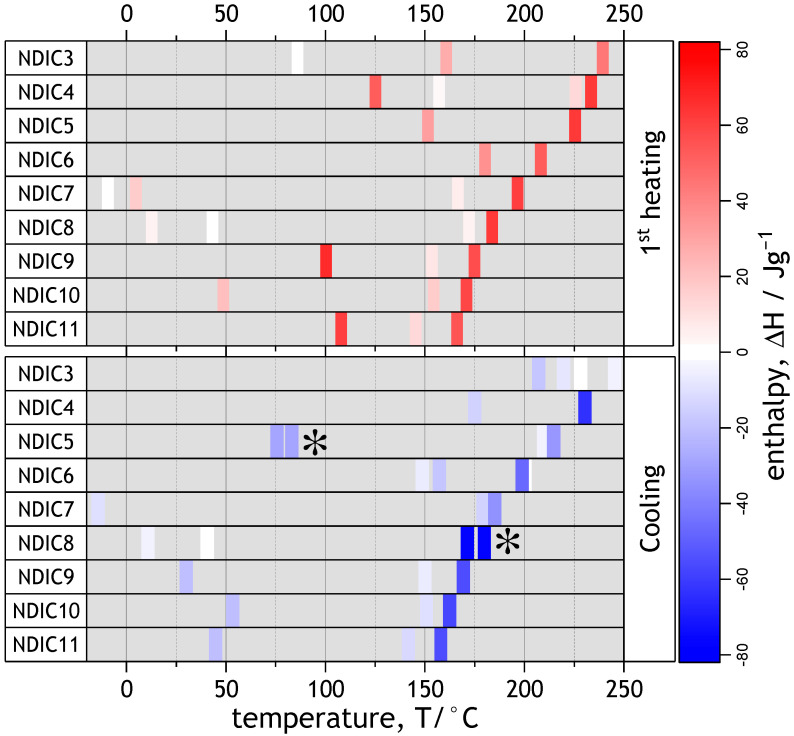
Transition points and corresponding enthalpies of NDIs measured with DSC during heating and cooling runs. Color indicates positive (red) or negative (blue) transition enthalpy. The shade indicates the magnitude of the enthalpy. Multi-peak transitions for which overall enthalpy has been calculated are indicated with asterisks. Original DSC plots are given in the Appendix A.

**Table 1 molecules-28-02940-t001:** Isolated yields of the synthesized NDIs.

	Yield [%] (15 mmol Scale)	Amine Boiling Point [°C]
NDIC3	35	47.8
NDIC4	51	78.0
NDIC5	88	104
NDIC6	82	131
NDIC7	75	155
NDIC8	85	176
NDIC9	78	200
NDIC10	79	217
NDIC11	68	242

**Table 2 molecules-28-02940-t002:** Crystallographic data of NDIs determined from single crystal and powder X-ray diffraction patterns. Unit cell parameters determined from the powder patterns are given in brackets. The explanation for why d_π-π_ for NDIC3 was not calculated is given in the text.

	Crystal System	Space Group	a	B	C	α	β	γ	d_π-π_ [Å]
NDIC3	orthorhombic	Pbca	6.96	17.24	27.58	90.0	90.0	90.0	n/a
			(7.15)	(17.49)	(27.80)	(90.0)	(90.0)	(90.0)	
NDIC4	triclinic	P$\overset{-}{1}$	5.22	7.84	11.13	103.7	94.3	93.9	3.34
			(5.32)	(8.17)	(11.04)	(105.1)	(93.4)	(92.9)	
NDIC5	monoclinic	P2_1_/c	5.03	8.11	24.21	90.0	90.8	90.0	3.22
			(5.12)	(7.94)	(26.24)	(90.0)	(91.6)	(90.0)	
NDIC6	triclinic	P$\overset{-}{1}$	4.90	8.28	14.52	96.3	98.1	93.6	3.32
			(4.91)	(8.38)	(14.55)	(96.0)	(99.1)	(93.1)	
NDIC7	monoclinic	P2_1_/c	7.87	4.84	33.02	90.0	92.0	90.0	3.27
			(-)	(-)	(-)	(-)	(-)	(-)	
NDIC8	triclinic	P$\overset{-}{1}$	4.77	6.53	22.71	87.9	88.9	75.8	3.36
			(4.86)	(6.63)	(22.87)	(88.7)	(89.9)	(75.3)	
NDIC9	monoclinic	P2_1_/c	7.85	4.84	37.74	90.0	95.0	90.0	3.17
			(7.95)	(4.91)	(38.06)	(90.0)	(94.9)	(90.0)	
NDIC10	triclinic	P$\overset{-}{1}$	4.73	6.55	25.58	94.2	95.3	104.3	3.31
			(4.81)	(6.70)	(26.09)	(93.2)	(95.8)	(105.2)	
NDIC11	monoclinic	P2_1_/c	7.85	4.87	42.51	90.0	92.1	90.0	3.14
			(-)	(-)	(-)	(-)	(-)	(-)	

**Table 3 molecules-28-02940-t003:** Electron mobilities and threshold voltages of NDI-based OFETs.

	Electron Mobility—μ_e_/10^−4^ × cm^2^V^−1^s^−1^	Threshold Voltage—V_th_/V
NDIC3	0.16 ± 0.03	41 ± 3
NDIC4	0.55 ± 0.13	34 ± 2
NDIC5	0.12 ± 0.01	46 ± 1
NDIC6	1.55 ± 0.31	37 ± 1
NDIC7	2.49 ± 0.79	47 ± 6
NDIC8	1.24 ± 0.46	47 ± 3
NDIC9	0.15 ± 0.03	46 ± 3
NDIC10	0.08 ± 0.01	54 ± 3
NDIC11	0.56 ± 0.05	46 ± 2

## Data Availability

Not applicable.

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
