# Peer review of "Synthesis, Solution, and Solid State Properties of Homological Dialkylated Naphthalene Diimides—A Systematic Review of Molecules for Next-Generation Organic Electronics"

_molecules, 2023, doi:10.3390/molecules28072940_

Round 1
Reviewer 1 Report
The title of the manuscript submitted by Kiersnowski and colleagues "Synthesis, Solution and Solid State Properties of Homological Dialkylated Naphthalene Diimides. A Systematic Review of Molecules for Next-Generation Organic Electronics " resumes very well the work described in it. The authors have improved the synthetic protocol for the preparation of these compounds, with a two-fold increase of the typical reported scale. On the other hand, although the studied compounds have been previously reported, their systematic and thorough characterization (spectroscopic, structural, morphological, thermal and electronically) has certainly provided interesting insights on this family of compounds which, to my point of view, merits publication in this journal.
In particular, the different characterization studies have been conceived based on the interest of these compounds in the field of organic electronics. For instance, the authors have investigated the solubility trends in different solvents, the crystalline structure and the pi-pi stacking distances, film formation features and field-effect electron mobilities. From their results, the authors propose a classification of the NDIs as aromatic- or alkyl-dominated NDI together with an intermediate type (transition NDIs). However, it is not very clear to me this classification respect to the solubility or the electron mobility properties, and even less respect to the IR and crystallographic properties, where odd-parity effects from the alkyl chain have been unveiled. Therefore, the authors should better explain the grounds for this classification so that the properties associated with a certain type of NDI are clearly identified.
Another major concern deals with the solubility studies. In paragraph 2.5 it is explained how the concentration–absorbance calibration curves were prepared. Which criteria were used for the selection of the absorption wavelength? Is the Beer-Lambert law still valid for the saturated solutions? Can the formation of aggregates be precluded for these compounds?
Other minor issues to be addressed:
- Page 3, section 2.1: information on the stationary phase for the chromatography is missing, as well as for TLC if used.
- Page 3, section 2.2: Fig. 2 should be placed in section 3.1 of the “results and discussion part”. In this section, on the contrary, the authors could include the general synthetic protocol from the supporting information. Moreover, they should additionally provide, for each compound, the yield and the most important characterization data by, at least, two different techniques (NMR, HRMS, elemental analysis…). In the current version of the manuscript, some of this information is distributed throughout the article, making it difficult to analyze.
- Page 6, from line 236 on: the role of the imidazole is not stated. In my opinion, it has more of a catalytic role for the amide bond formation than a solvating media.
- Page 7: data on the solubility in CHCl3 is missing in the main article (and SI) while being described in the text.
- Pages 8–10: tables 2 and 3 could be transferred to the SI.
- Page 11, Fig. 5: “a”, “b”, “c” and “d” are missing.
- Page 11, line 390: “numbers in parenthesis” instead of “numbers in brackets”.
- Page 12, lines 398 and 399: stacking distances are in Å, not in nm.
- Page 13: Fig. 6 could be reformatted so as to better occupy the space.
- Page 14, lines 438–440: the deviation of NDIC7’s solubility is only observed in heptane, not in toluene or ortho-dichlorobenzene.
- Table S1: the described molar absorptivity values are not epsilon∙103 but epsilon∙10-3 instead.
Author Response
We are grateful to the Reviewer#1 for the effort put into evaluating our paper. We appreciate the insightful analysis of our manuscript. The manuscript has been improved based on the comments. Below, we provide our replies to comments and explain the way the manuscript was modified in a "point-by-point" manner . We hope that the extent of our correction will fulfill the expectations of both the Reviewer as well as the Editor.
Reviewer #1.1: However, it is not very clear to me this classification respect to the solubility or the electron mobility properties, and even less respect to the IR and crystallographic properties, where odd-parity effects from the alkyl chain have been unveiled. Therefore, the authors should better explain the grounds for this classification so that the properties associated with a certain type of NDI are clearly identified.
Authors to #1.1: We are particularly grateful for this remark. We have rediscussed this point and, after a thorough consideration, we decided to quit the initial idea of classification and rephrase corresponding parts of the manuscript. We agree with the Reviewer that there is actually no clear-cut ground for the classification proposed in the initial version of the manuscript. Appropriate changes are included in the revised introduction, conclusions, and the abstract.
Reviewer #1.2: In paragraph 2.5 it is explained how the concentration–absorbance calibration curves were prepared. Which criteria were used for the selection of the absorption wavelength? Is the Beer-Lambert law still valid for the saturated solutions? Can the formation of aggregates be precluded for these compounds?
Authors to #1.2: The choice of absorption wavelengths for solubility studies was based on technical possibility to measure the full spectrum of a saturated solution. In other words, for the spectra where the absorption was lower than 2.5 (the spectrometer sensitivity limit), the wavelenght of absorption peak was chosen. In the case of the samples/spectra where absorption exceeded 2.5 the calculations were based on the data from the edge of absorption peak, taking the wavelength as close to peak maximum as possible (i.e. the one still measurable at high concentration, with A<2.5). The validity of Lambert-Beer law was verified by preparation of several solutions with concentrations close to the solubility limit. As they were following the calibration line and, additionally, all absorption spectrum features were preserved there was no sign of formation of aggregates. Appropriate information has been added to the manuscript at the very end of paragraph 2.5.
Reviewer #1.3: information on the stationary phase for the chromatography is missing, as well as for TLC if used.
Authors to #1.3: A suitable information has been added in the revised manuscript. The whole synthetic part of the experimental section has been rewritten according to the suggestion included in the Reviewer#1.4 remark. TLC was not used in the study.
Reviewer #1.4: Page 3, section 2.2: Fig. 2 should be placed in section 3.1 of the “results and discussion part”. In this section, on the contrary, the authors could include the general synthetic protocol from the supporting information. Moreover, they should additionally provide, for each compound, the yield and the most important characterization data by, at least, two different techniques (NMR, HRMS, elemental analysis…). In the current version of the manuscript, some of this information is distributed throughout the article, making it difficult to analyze.
Authors to #1.4: The manuscript has been revised as suggested by the Reviewer. The whole synthetic protocol is included in the main part of the manuscript (the details have been transferred from the SI). The spectroscopic identification data is now provided in the classical form as sugested by the Reviewer.
Reviewer #1.5: Page 6, from line 236 on: the role of the imidazole is not stated. In my opinion, it has more of a catalytic role for the amide bond formation than a solvating media.
Authors to #1.5: The extended discussion of the imidazole is included in the revised section 3.1. Based on the survey of literature it is, however, difficult to ultimately define the imidazole's role. Some observations - in both our study, as well as previously reported in literature (e.g. 10.1002/cber.19851181138, 10.1002/hlca.200590225, ), point towards its catalytic role, which is mentioned in the updated discussion. More trivially, however, imidazole can also exert an influence on the reaction by increasing solubility of the anhydride. Since, based on our results we are unable to indicate which of these effects has stronger influence on the reaction we have left this part of discussion without a clear conclusion.
Reviewer #1.6: Page 7: data on the solubility in CHCl3 is missing in the main article (and SI) while being described in the text.
Authors to #1.6: That is true. Basically, chloroform is a known good solvent for NDIs and other rylene dyes. Our measurements indicated that solubility of most of NDIs in chloroform oscillates around or exceeds 50 mg/mL. Such concentration is beyond levels typically used in fabrication of organic electronic devices and therefore strict quantitative data is in such a case less important. Hence we just mention the data for chloroform, and the discussion is focused on heptane, toluene and dichlorobenzene, that are less obvious, yet still important solvents of NDIs . Hence, no changes were made to the manuscript based on this remark. We hope that the Reviewer can kindly accept that.
Reviewer #1.7 Pages 8–10: tables 2 and 3 could be transferred to the SI.
Authors to #1.7: Since the manuscript has been modified as mentioned in point 1.4, the tables were no longer necessary and they were transferred to the SI, just as suggested by the Reviewer.
Reviewer #1.8: Page 11, Fig. 5: “a”, “b”, “c” and “d” are missing.
Authors to #1.8: Corrected.
Reviewer #1.9: Page 11, line 390: “numbers in parenthesis” instead of “numbers in brackets”.
Authors to #1.9: Changed as suggested.
Reviewer #1.10: Page 12, lines 398 and 399: stacking distances are in Å, not in nm.
Authors to #1.10: Corrected.
Reviewer #1.11: Page 13: Fig. 6 could be reformatted so as to better occupy the space.
Authors to #1.11: The micrographs in revised Figure 6 were clustered in the different way to save the space.
Reviewer #1.12: Page 14, lines 438–440: the deviation of NDIC7’s solubility is only observed in heptane, not in toluene or ortho-dichlorobenzene.
Authors to #1.12: The statement was made more specific, as pointed by the Reviewer.
Reviewer #1.13: Table S1: the described molar absorptivity values are not epsilon∙103 but epsilon∙10-3 instead.
Authors to #1.13: Corrected.
Reviewer 2 Report
Manuscript ID: molecules-2285715
Title: Synthesis, Solution and Solid State Properties of Homological Dialkylated Naphthalene Diimides. A Systematic Review of Molecules for Next-Generation Organic Electronics.
Authors: Dorota Chlebosz, Waldemar Goldeman, Krzysztof Janus, Michał Szuster, Adam Kiersnowski *
The present work is a systematic review of symmetrical N, N’-dialkylated naphthalene diimines (NDIs). The authors through an experimental study extensively referred to literature-reported data, and present a correlation between the molecular structure, solubility, self-assembly, and electronic properties of a series of NDIs with increasing alkyl chain length. The NDIs were synthesized according to literature-reported routes and were characterized through NMR, UV-vis, FT-IR, TGA-DSC, XRD, and polarized optical spectroscopy. They were employed in organic field effect transistors (OFETs) to study their electron field mobilities. The dependence of the NDIs' properties on the length of their alkyl chain was deeply discussed.
The present systematic review is well-organized, and all the sections are well-balanced. The treated arguments are well described, and the experimental data are supported by adequate references. I suggest it for publication on Molecules only after some minor revisions.
Suggested remarks:
- Figure 1: the acronyms of the samples should be explained in the main text, and not only in the figure caption.
- Tables 2 and 3 are not clear, the authors are invited to improve the graphic.
- Page 7 line 268: mind the point.
Author Response
We are grateful to the Reviewer#2 for evaluating our paper. We were happy to see the very positive reception of our work. We have improved the manuscript based on the comments. The details of the correction are provided below.
Reviewer #2.1: Figure 1: the acronyms of the samples should be explained in the main text, and not only in the figure caption.
Authors to #2.1: The acronyms of the samples are now explained already in the experimental section, where, additionally systematic IUPAC names were provided.
Reviewer #2.2: Tables 2 and 3 are not clear, the authors are invited to improve the graphic.
Authors to #2.1: Based on the comments from Reviewer#1, the tables were moved to the supporting information (SI). In the SI, they were rotated so that the data are clearly visible.
Reviewer #2.3: Page 7 line 268: mind the point.
Authors to #2.3: The unnecessary full stop has been deleted